# Reducing Transformer Key-Value Cache Size with Cross-Layer Attention

**William Brandon**[*]
MIT CSAIL
wbrandon@csail.mit.edu

**Mayank Mishra**[*]
MIT-IBM Watson AI Lab

**Aniruddha Nrusimha**
MIT CSAIL

**Rameswar Panda**
MIT-IBM Watson AI Lab

**Jonathan Ragan-Kelley**
MIT CSAIL

## Abstract

Key-value (KV) caching plays an essential role in accelerating decoding for transformer-based autoregressive large language models (LLMs). However, the amount of memory required to store the KV cache can become prohibitive at long sequence lengths and large batch sizes. Since the invention of the transformer, two of the most effective interventions discovered for reducing the size of the KV cache have been Multi-Query Attention (MQA) and its generalization, Grouped-Query Attention (GQA). MQA and GQA both modify the design of the attention block so that multiple query heads can share a single key/value head, reducing the number of distinct key/value heads by a large factor while only minimally degrading accuracy. In this paper, we show that it is possible to take Multi-Query Attention a step further by also sharing key and value heads between adjacent layers, yielding a new attention design we call Cross-Layer Attention (CLA). With CLA, we find that it is possible to reduce the size of the KV cache by another $2\times$ while maintaining nearly the same accuracy as unmodified MQA. In experiments training 1B- and 3B-parameter models from scratch, we demonstrate that CLA provides a Pareto improvement over the memory/accuracy tradeoffs which are possible with traditional MQA, potentially enabling future models to operate at longer sequence lengths and larger batch sizes than would otherwise be possible.

## 1 Introduction

The memory footprint of the key-value (KV) cache can be a bottleneck when serving large language models (LLMs). Because the size of the KV cache scales proportionally with both sequence length and batch size, the memory overhead of KV cache storage can limit batch sizes when operating on long sequence lengths [Chowdhery et al., 2022], and can require employing costly techniques like offloading when on-device memory is scarce [Sheng et al., 2023]. It is also desirable to be able to persist KV caches over long periods of time in order to minimize redundant computations [Gao et al., 2024, Google, 2024]. However, the size of the KV cache directly determines the cost of storing and retrieving such persistent caches. As new applications of LLMs emerge which demand ever-longer sequence lengths, the memory footprint of the KV cache is becoming an increasingly important consideration in the design of efficient transformer-based language models.

Existing work has proposed a variety of methods for decreasing the memory footprint of the KV cache, including storing KV activations in low precision [Hooper et al., 2024, Zhang et al., 2024b],

---

[*]Equal Contribution

38th Conference on Neural Information Processing Systems (NeurIPS 2024).

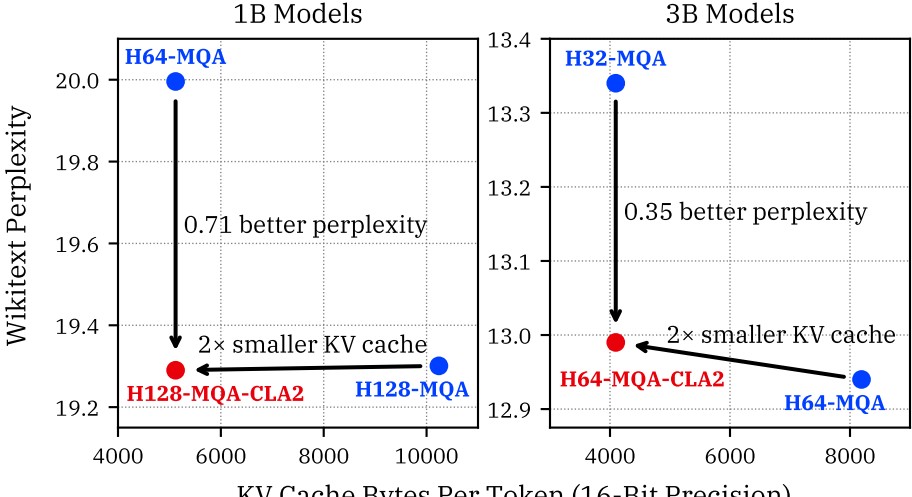

Figure 1: Accuracy/memory tradeoffs achieved by MQA models with CLA (red) and without CLA (blue) at the 1B-parameter and 3B-parameter scale, as measured by perplexity on Wikitext. We find that CLA provides the same reduction in KV cache size as shrinking the head dimension $d_{\text{head}}$ by $2\times$ while achieving substantially lower perplexities. More details on these experiments are presented in sections 3.2.2 and 3.3.

evicting unimportant KV cache entries [Zhang et al., 2023, Liu et al., 2023], and sharing keys and values across query heads in the attention mechanism [Shazeer, 2019, Ainslie et al., 2023].

In this paper, we introduce a method for reducing the size of the KV cache along a dimension different than those explored in prior work: namely, reducing the number of unique *layers* in the KV cache. Our contributions are as follows:

1. We propose *Cross-Layer Attention* (CLA), a modification to the transformer architecture which reduces the size of the KV cache by *sharing KV activations across layers*.

2. We conduct extensive pretraining experiments to characterize the effect of different CLA configurations on accuracy and memory usage across a range of architectural hyperparameters, learning rates and model sizes.

3. We demonstrate that CLA enables accuracy/memory Pareto improvements relative to existing Multi-Query Attention (MQA) and Grouped-Query Attention (GQA) architectures.

4. In particular, we demonstrate at the 1B- and 3B-parameter scales that combining CLA with MQA can achieve a $2\times$ reduction in KV cache size versus a plain MQA baseline, with minimal degradation in perplexity.

5. We offer guidance on which CLA configurations perform best based on our experiments, finding that CLA should be used between pairs of consecutive layers, and that CLA appears to deliver the most robust benefits when used in conjunction with MQA.

## 2 Cross-Layer Attention

In this section we describe our Cross-Layer Attention (CLA) technique, and its relationship to the KV-sharing mechanisms employed by the existing Multi-Query and Grouped-Query attention architectures (MQA and GQA).

### 2.1 Background: Multi-Query Attention and Grouped-Query Attention

The original transformer architecture employed Multi-Head Attention (MHA) [Vaswani et al., 2017], in which each query head attends over the keys and values produced by a distinct key/value head. In

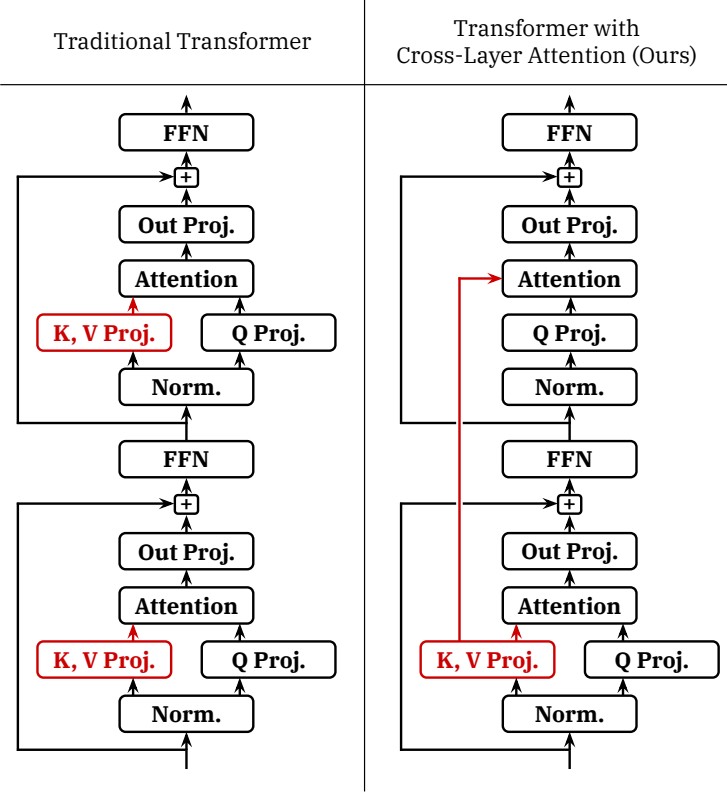

| Traditional Transformer | Transformer with Cross-Layer Attention (Ours) |

Figure 2: Schematic of two consecutive layers in a transformer using a traditional attention design (left) and in a transformer using Cross-Layer Attention (right). When using traditional attention, each layer computes its own separate $K$ and $V$ activations, which must be cached on a per-layer basis during autoregressive decoding. When using Cross-Layer Attention, some layers compute their own fresh $K$ and $V$ activations, while other layers reuse the $K$ and $V$ activations of earlier layers.

MHA, the KV activations of each key/value head must be stored separately in the KV cache, resulting in a storage overhead of $2 \cdot n_{\text{query}} \cdot d_{\text{head}}$ elements per token, where $n_{\text{query}}$ is the number of query heads and $d_{\text{head}}$ is the embedding dimension of each head.

To reduce the overhead associated with storing and accessing the KV cache during transformer decoding, Shazeer [2019] proposed Multi-Query Attention (MQA), which Ainslie et al. later generalized to Grouped-Query Attention (GQA). Grouped-Query Attention modifies the transformer architecture by organizing the query heads of each attention layer into groups, where each group of query heads shares a single key/value head. Because the size of the KV cache scales only with the number of distinct key/value heads, not the number of query heads, GQA reduces the storage overhead of the KV cache to $2 \cdot n_{\text{group}} \cdot d_{\text{head}}$, where $n_{\text{group}}$ denotes the number of groups for GQA and $n_{\text{group}} < n_{\text{query}}$. MQA can be seen as the special case of GQA in which $n_{\text{group}} = 1$.

Shazeer and Ainslie et al. find that MQA and GQA enable significant reductions in KV cache size and decoding latency while incurring only a small degradation in accuracy compared to MHA architectures with the same head dimension. The family of attention architectures enabled by using MQA and GQA defines an accuracy/memory tradeoff space in which model designers can choose how they want to balance the expressive power and KV cache overhead of their attention mechanism. MQA and GQA stake out different positions in this tradeoff space, and neither is necessarily preferable to the other for all use cases.

## 2.2 Sharing KV Activations Across Layers

Inspired by the success of MQA and GQA, which share key/value heads across query heads within a single layer, we propose also sharing key/value heads *across layers*. We refer to such an attention

architecture as *Cross-Layer Attention* (CLA), and present a diagrammatic view of it in Figure 2. CLA computes key/value projections for only a subset of layers in the model; the attention blocks in layers without key/value projections reuse the KV activations of previous layers. Only the subset of layers with key/value projections contribute to the KV cache, allowing a reduction in memory footprint relative to traditional architectures which apply a separate key/value projection in each layer.

CLA is orthogonal to MQA/GQA/MHA, and can be combined with any of them. Moreover, in the same way that GQA allows varying $n_{group}$ to access a family of different attention configurations, CLA allows varying the number of layers which share the output of each KV projection, which we refer to as the *sharing factor*. We refer to different configurations of CLA by their sharing factors, giving rise to CLA2, which shares each KV projection among a pair of adjacent layers, CLA3, which shares each KV projection among a group of 3 layers, and so on. In Appendix A we include a figure illustrating a few of the different attention configurations possible with CLA.

### 2.3 Implications for System Design

CLA is primarily an intervention to reduce the memory footprint of the KV cache, and only has minor effects on other resources consumed by the model during training and inference. Here, we summarize the effect of CLA on key metrics relevant from a systems engineering perspective, assuming all other architectural hyperparameters are held constant:

- **KV Cache Memory**: CLA significantly reduces KV cache memory footprint, shrinking it by a factor equal to the sharing factor, or slightly less if the sharing factor does not evenly divide the number of layers.

- **Training Memory Footprint**: CLA reduces the memory footprint of intermediate KV activation tensors materialized during training, although for GQA and MQA models such KV tensors are typically small compared to the model's hidden states and MLP activations.

- **Model Parallelism**: CLA is fully compatible with standard tensor parallelism techniques [Shoeybi et al., 2020] for sharding model weights across multiple accelerators. In the presence of pipeline parallelism [Huang et al., 2019], either different layers which share a KV cache must be kept in the same pipeline stage, or else KV activations must be communicated between pipeline stages.

- **Parameters and FLOPs**: Because CLA reduces the total number of key/value projection blocks in the model, CLA slightly reduces the number of parameters in the model and the number of FLOPs required during a forward or backward pass.

- **Decoding Latency**: In the context of a full LLM serving stack, CLA can enable larger batch sizes and longer KV cache persistence times than would otherwise be possible, which have the potential to improve inference latency.

- **Core Attention Latency**: Unlike MQA and GQA, CLA has no direct effect on the memory bandwidth consumed by the attention mechanism in each decoding step, because even shared KV cache layers must be separately re-read from main memory in each attention layer. CLA therefore has no direct effect on the latency of the core attention computation during decoding.

## 3 Pretraining Experiments

To determine the effect of Cross-Layer Attention on language modeling accuracy, we trained a collection of transformer-based language models from scratch at the 1 billion and 3 billion parameter scales. While running these experiments, we sought to answer the following questions:

1. What accuracy/memory tradeoffs are possible using CLA?
2. How does using CLA compare to using plain GQA or MQA?
3. How does CLA interact with GQA and MQA?
4. What CLA configurations perform best given a fixed memory budget?
5. Are the effects of CLA consistent across scales?

| Model | $d_{\text{head}}$ | Query Heads | KV Heads | KV Layers | KV Bytes Per Token (16-Bit) | Validation Perplexity |
|---|---|---|---|---|---|---|
| **Non-CLA Baselines** | | | | | | |
| H128-MHA | 128 | 16 | 16 | 20 | 163 840 | 13.15 |
| H128-GQA4 | 128 | 16 | 4 | 20 | 40 960 | 13.36 |
| H128-GQA2 | 128 | 16 | 2 | 20 | 20 480 | 13.52 |
| H128-MQA | 128 | 16 | 1 | 20 | 10 240 | 13.54 |
| H64-MQA | 64 | 32 | 1 | 20 | 5120 | 13.81 |
| H46-MQA | 46 | 45 | 1 | 20 | 3680 | 13.96 |
| H32-MQA | 32 | 64 | 1 | 20 | 2560 | 14.37 |
| **MQA + CLA2 Models** | | | | | | |
| H512-MQA-CLA2 | 512 | 4 | 1 | 10 | 20 480 | 13.49 |
| H256-MQA-CLA2 | 256 | 8 | 1 | 10 | 10 240 | 13.51 |
| H128-MQA-CLA2 | 128 | 16 | 1 | 10 | 5120 | 13.60 |
| H90-MQA-CLA2 | 90 | 22 | 1 | 10 | 3600 | 13.73 |
| H64-MQA-CLA2 | 64 | 32 | 1 | 10 | 2560 | 13.89 |

Table 1: Results of our 1B-scale design space exploration. Full results including ablations can be found in Appendix B.

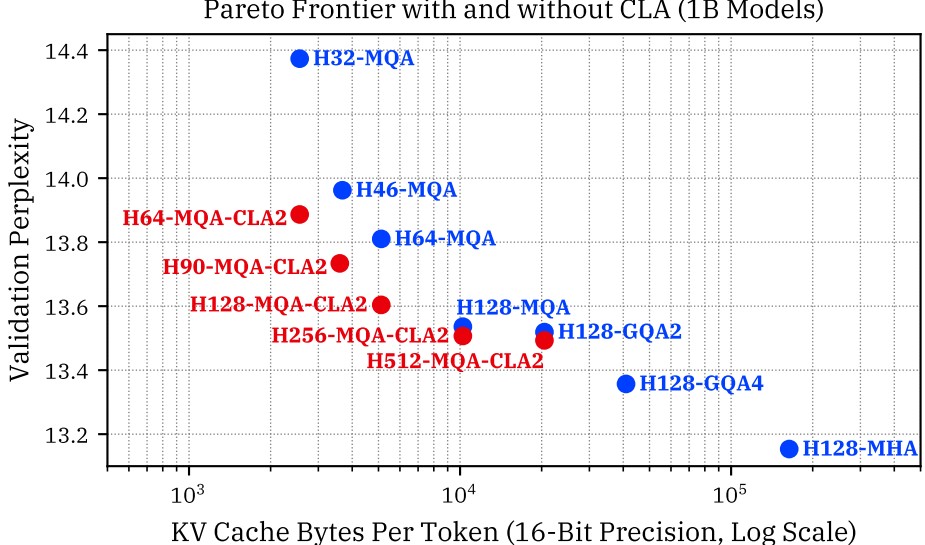

Figure 3: The accuracy/memory Pareto frontier discovered in our 1B-scale design space exploration, for models with CLA (red) and without CLA (blue). Lower is better on both axes.

We found that CLA enables favorable accuracy/memory tradeoffs compared to what is possible using plain GQA or MQA. Moreover, we found that in our experimental regime, a sharing factor of 2 is more effective than other sharing factors, and that CLA is consistently effective when combined with MQA when trying to decrease KV cache storage. Finally, we found that CLA confers benefits at both 1B- and 3B-parameter scales. In the rest of this section, we present our experimental setup and results in more detail.

### 3.1 Common Experimental Parameters

In all our experiments, we train our models from scratch on data from the SlimPajama [Soboleva et al., 2023] dataset, tokenized with the GPT-NeoX tokenizer [Black et al., 2022] which uses Byte-Pair Encoding (BPE) [Wang et al., 2019]. We adopt a Llama-like [Touvron et al., 2023] architecture

| Model Family | Hidden Size | FFN Size | Layers | Sequence Length | Training Tokens |
|---|---|---|---|---|---|
| 1B Models | 2048 | 5472 | 20 | 2048 | $\approx 30 \times 10^9$ |
| 3B Models | 3072 | 8192 | 32 | 2048 | $\approx 100 \times 10^9$ |

Table 2: Architectural and training hyperparameters shared across our pretraining experiments.

with pre-normalization [Xiong et al., 2020], SwiGLU activations [Shazeer, 2020, Ramachandran et al., 2017], and rotary position embeddings [Su et al., 2023]. We do not use dropout for any of our models. Our models use RMSNorm [Zhang and Sennrich, 2019] with learnable elementwise multiplication parameters, and our CLA models use separately-learnable RMSNorm parameters for the KV projection blocks and Q projection blocks in attention. Unless otherwise stated, we always set the number of query heads $n_{query}$ such that $n_{query} \cdot d_{head}$ is equal to the hidden size $d_{model}$.

We train all models using the AdamW optimizer [Loshchilov and Hutter, 2019] with gradient clipping, using $\beta_1 = 0.9$, $\beta_2 = 0.95$, a weight decay factor of $0.1$, and a clipping norm of $1.0$. We use a linear learning rate warmup for the first $5\%$ of training examples and a cosine learning rate schedule Loshchilov and Hutter [2017] decaying to $10\%$ of the peak learning rate over the remainder of training. We set the sequence length to 2048 tokens and the batch size to 2048 sequences, for a total of $\approx 4M$ tokens per training step. All our experiments initialize the weights of linear layers from a normal distribution with mean zero and standard deviation 0.01275.

We perform all experiments on NVIDIA H100 GPUs using PyTorch [Paszke et al., 2019, Ansel et al., 2024]. We use mixed precision training [Micikevicius et al., 2018] in BF16 [Kalamkar et al., 2019] with gradient all-reduce and gradient accumulation in FP32 for training stability.

## 3.2 Experiments at 1B-Parameter Scale

We trained all our 1B-scale models on 30 billion tokens using a consistent data order, and, other than varying the attention mechanism, used the same architectural hyperparameters across all 1B-scale models. This means that all our 1B models were all trained using approximately the same number of FLOPs and approximately the same number of GPU-hours, with CLA models requiring slightly fewer FLOPs to train than their non-CLA counterparts due to the reduced number of key/value projections. The common hyperparameters shared across our 1B-scale experiments can be found in Table 2.

We ran two main sets of experiments at the 1B-parameter scale. First, we trained a diverse set of CLA and non-CLA models to characterize the range of accuracy/memory tradeoffs achievable with and without CLA, and to determine which CLA configurations are most effective; we refer to these as our *design space exploration* experiments, and describe them in more detail in Section 3.2.1. Second, we conducted a learning rate sweep on a subset of models from our design space exploration to verify that our results continue to hold even against a strong non-CLA baseline with a well-tuned learning rate. We describe these learning rate tuning experiments in Section 3.2.2.

### 3.2.1 Design Space Exploration

The primary goal of our 1B-parameter-scale design space exploration was to characterize the Pareto frontier of accuracy/memory tradeoffs achievable with and without CLA, and to determine which CLA configurations achieve the best accuracy on a fixed KV cache memory budget. We train all models in our design space exploration using a learning rate of $LR = 3 \times 10^{-4}$, which we determined to be conservative; we explore the effect of the learning rate on accuracy in more detail in Section 3.2.2.

For our design space exploration, we first trained a collection of seven non-CLA baseline models along the MHA-GQA-MQA spectrum, exhibiting a range of KV cache memory requirements spanning two orders of magnitude. Our baseline model with the largest KV cache memory footprint is an MHA model with a head embedding dimension of $d_{head} = 128$ (163840 bytes per token at 16-bit precision), and our baseline with the smallest footprint is an MQA model with head dimension $d_{head} = 32$ (2560 bytes per token).

We quantify the accuracy of models in our design space exploration using perplexity on a held-out validation set of $\approx 4M$ tokens drawn from our SlimPajama corpus. A summary of results for the

models in our design space exploration, including our baseline models, can be found in Table 1. We adopt the naming scheme "H⟨$d_{\text{head}}$⟩-⟨attention mechanism⟩" for all models in our experiments, so that, for example, a model employing MQA with a head dimension of $d_{\text{head}} = 64$ would be named "H64-MQA." For our baseline models, we observe that validation perplexity increases monotonically as we reduce the memory capacity of the KV cache, ranging from a perplexity of 13.15 for our H128-MHA baseline to 14.37 for our H32-MQA baseline.

In the rest of this section, we present results for the CLA models we trained during our design space exploration.

**Best Performance: MQA + CLA2.** We trained a family of five models combining MQA with CLA2. We varied the head dimension for our MQA-CLA2 models from $d_{\text{head}} = 512$ down to $d_{\text{head}} = 64$, allowing us to compare to a range of non-CLA baseline models with varying KV cache capacities.

We found that our MQA-CLA2 models are able to achieve better perplexities than baseline models requiring the same amount of KV cache memory, advancing the accuracy/memory Pareto frontier. We present a plot of the accuracy/memory Pareto frontier with and without CLA in Figure 3. Our MQA-CLA2 models with head dimensions $d_{\text{head}} \in \{64, 90, 128\}$ are able to match the KV cache memory footprint of baseline MQA models with head dimensions $d_{\text{head}} \in \{32, 46, 64\}$ while achieving substantial perplexity improvements in the range of 0.21–0.48 points. Additionally, our MQA-CLA2 models with large head sizes of $d_{\text{head}} \in \{256, 512\}$ are able to match the KV cache footprint of our MQA and GQA2 baselines with $d_{\text{head}} = 128$ while achieving a small perplexity improvement of 0.03 points.

**Ablations.** We found that our MQA-CLA2 models achieved the best accuracy/memory tradeoffs among all CLA configurations we tested in our design space exploration. Here, we briefly describe the ablations we conducted to explore alternate CLA configurations. We present our ablations in more detail in Appendix B.

We explored combining CLA with GQA4 and GQA2, and found that all our experiments combining CLA with GQA either matched the accuracy/memory tradeoffs of our MQA-CLA2 models, or underperformed our non-CLA baselines. We trained models using CLA3 and CLA4, and found that they achieved accuracy/memory Pareto improvements over our non-CLA baselines, but slightly underperformed our MQA-CLA2 models at the same memory budget. Finally, we explored more complex and irregular sharing patterns than sharing a single KV cache across each pair of adjacent layers, but found no benefit over our basic CLA2 configuration from any of the patterns we tested.

### 3.2.2 Robustness to Learning Rate Tuning

The relative performance of different model architectures can change depending on the learning rates at which they are evaluated. To account for the effects of the learning rate on our results, we conducted learning rate tuning experiments on three models of interest from our initial 1B-scale design space exploration. These learning rate tuning experiments help us verify that CLA continues to provide benefits even when compared to baselines trained at their optimal learning rates.

We chose to tune the learning rate for the baseline models H128-MQA and H64-MQA, as well as the CLA model H128-MQA-CLA2. In our initial design space exploration, our results for these models indicated that CLA makes it possible to shrink the KV cache footprint of an MQA model with $d_{\text{head}} = 128$ by a factor of $2\times$ while incurring only a small (0.06 point) degradation in perplexity, or to create a model with the same KV cache footprint as an MQA model with $d_{\text{head}} = 64$ while enjoying a substantial (0.21 point) improvement in perplexity. We wanted to verify that this qualitative pattern continues to hold when all models are trained with well-tuned learning rates.

**Learning Rate Tuning Strategy.** For each of our three model configurations, we swept the learning rate upwards from an initial value of $3 \times 10^{-4}$ in multiplicative increments of $1.5\times$. We ended our sweep for each model at the point where validation perplexity stopped improving.

**Results.** We found an optimal learning rate of LR $= 1.5 \times 10^{-3}$ for our H128-MQA baseline, and a higher optimal learning rate of LR $= 2.25 \times 10^{-3}$ for both our H64-MQA baseline and our H128-MQA-CLA2 model.

The results of our 1B-scale learning rate tuning experiments can be found in Table 3. When comparing all three models at their best learning rates, we found that the qualitative result from our design space exploration continues to hold: our CLA2 model incurs only a small (0.04 point) validation perplexity degradation relative to our $d_{\text{head}} = 128$ baseline while enjoying a $2\times$ smaller KV cache footprint, and achieves a substantial (0.31 point) validation perplexity improvement compared to our $d_{\text{head}} = 64$ baseline while using the same amount of KV cache memory.

To further validate our results, we also evaluate our three learning-rate-tuned 1B-scale models under EleutherAI's LM Eval Harness [Gao et al., 2023] on Wikitext [Merity et al., 2017] perplexity and seven standard downstream benchmarks. On Wikitext perplexity, we observe a similar pattern as with validation perplexity. On the downstream evaluations, we found that none of our three models model consistently wins or loses across different benchmarks, and that all three models are consistently within 1–5 percentage points of each other.

| Model | KV Bytes Per Token (16-bit) | Best LR | Validation Perplexity | Wikitext Perplexity |
|---|---|---|---|---|
| H128-MQA | 10240 | $1.5 \ \times 10^{-3}$ | **12.39** | 19.30 |
| H128-MQA-CLA2 | 5120 | $2.25 \times 10^{-3}$ | 12.43 | **19.29** |
| H64-MQA | 5120 | $2.25 \times 10^{-3}$ | 12.74 | 20.00 |

| Model (Best LR) | Hellaswag | PIQA | WG | SciQ | OBQA | BoolQ | ARC-E |
|---|---|---|---|---|---|---|---|
| H128-MQA | **36.24** | 69.15 | **52.96** | **82.9** | 19.0 | **57.40** | **55.43** |
| H128-MQA-CLA2 | 36.01 | 69.15 | 51.93 | 82.6 | **21.4** | 53.21 | 53.87 |
| H64-MQA | 35.22 | **69.21** | 50.75 | 78.5 | 19.4 | 55.81 | 51.68 |

Table 3: Results of our learning rate tuning experiments at 1B scale. The columns "WG" and "OBQA" denote "WinoGrande" and "OpenBookQA", respectively.

## 3.3 Experiments at 3B-Parameter Scale

To determine how CLA performs when applied to larger models, we trained a collection of models at the 3B-parameter scale both with and without CLA. We trained each of our 3B-scale models from scratch on approximately 100B tokens from our SlimPajama corpus. The common architectural hyperparameters for our 3B-scale models can be found in Table 2.

**Initial 3B-Scale Experiments.** We initially tried training a 3B-parameter MQA model with $d_{\text{head}} = 128$ as one of the baselines for our experiments. To our surprise, we found that an MQA model with $d_{\text{head}} = 64$ outperformed this $d_{\text{head}} = 128$ baseline in perplexity despite using only $1/2$ as much KV cache memory. We also found that an MQA-CLA2 model with $d_{\text{head}} = 128$ was able to outperform both plain MQA baselines given a well-tuned learning rate. After seeing these results, we chose to use an MQA model with $d_{\text{head}} = 64$ as our main large-KV-cache baseline for further 3B-scale experiments, in order to ensure we were comparing to the strongest baseline we could identify. We provide more detail on these initial 3B-scale experiments in Appendix E.

| Model | KV Bytes Per Token (16-bit) | Best LR | Wikitext Perplexity |
|---|---|---|---|
| H64-MQA | 8192 | $1.0 \times 10^{-3}$ | **12.94** |
| H64-MQA-CLA2 | 4096 | $1.0 \times 10^{-3}$ | 12.99 |
| H32-MQA | 4096 | $1.0 \times 10^{-3}$ | 13.34 |

| Model (Best LR) | Hellaswag | PIQA | WG | SciQ | OBQA | BoolQ | ARC-E |
|---|---|---|---|---|---|---|---|
| H64-MQA | **47.34** | **74.54** | **60.46** | **88.9** | 24.2 | 57.25 | **66.92** |
| H64-MQA-CLA2 | 47.32 | **74.54** | 57.46 | 87.9 | 25.2 | 61.62 | 65.53 |
| H32-MQA | 46.05 | 73.83 | 60.06 | 88.6 | **25.6** | **61.87** | 65.24 |

Table 4: Results for our main 3B-scale experiments.

**Main 3B-Scale Experiments.** In our main 3B-scale experiments, we chose to compare an MQA-CLA2 model with $d_{\text{head}} = 64$ to a plain MQA model with $d_{\text{head}} = 64$, and to a plain MQA model with $d_{\text{head}} = 32$. We trained all models for our main experiments with a learning rate of LR $= 10^{-3}$, which we found to be optimal for our $d_{\text{head}} = 64$ MQA baseline model in our initial experiments. For our $d_{\text{head}} = 64$ MQA-CLA2 model and our $d_{\text{head}} = 32$ MQA baseline model, we also experimented with learning rates of LR $\in \{6.75 \times 10^{-4}, 1.5 \times 10^{-3}\}$, but found these achieved worse perplexities than our initial value of LR $= 10^{-3}$.

We report perplexity and downstream benchmark results for our main 3B experiments in Table 9. In the Wikitext perplexity results for this set of experiments, we find agreement with the pattern observed at the 1B scale. Our MQA-CLA2 model with $d_{\text{head}} = 64$ incurs only a small (0.05 point) degradation in perplexity compared to our $d_{\text{head}} = 64$ baseline while enjoying a $2\times$ smaller KV cache footprint, and achieves a substantial (0.35 point) improvement in perplexity compared to our $d_{\text{head}} = 32$ baseline while using the same amount of KV cache memory.

We also evaluate these three models on downstream benchmarks, and report the results in Table 4. As with our downstream benchmark evaluations at 1B-parameter scale, we find that all models perform similarly as measured by these downstream evaluations.

## 3.4 Comparison to Open Model

We wanted to verify that models trained using CLA remain competitive when compared with external models not trained using our particular software stack. To verify this, we ran an experiment which compares directly against the open-source GQA4 model TinyLlama-1.1B [Zhang et al., 2024a] at its 105B-token intermediate checkpoint. For this experiment, we pretrained our own version of TinyLlama-1.1B-105B from scratch using CLA2, using otherwise-identical training data, model architecture, and hyperparameters as described in the TinyLlama repository.

In this comparison, we found that our TinyLlama-1.1B-105B checkpoint trained with CLA2 matches or exceeds the performance of the original, publicly-available TinyLlama-1.1B-105B checkpoint. Due to space constraints, we present the full results of this comparison in Appendix G.

## 4 Adaptation Experiments

Ainslie et al. [2023] propose a recipe for converting MHA models into GQA models via a small amount of additional training, which they refer to as "uptraining." Inspired by Ainslie et al., we conducted experiments to investigate if models pretrained without CLA can be adapted to use CLA.

**Methods.** At the 1B- and 3B-parameter scales, we trained two "base" models on 105B tokens each, both using plain MQA. We then used the final weights of each base model to initialize an "adapted" model employing CLA2, which we trained on a further 21B tokens. More details on the hyperparameters used for these experiments can be found in Appendix H.1.

To initialize the KV projection weights for each group of layers sharing a KV cache in the adapted model, we considered two strategies: *copying* the base-model KV projection weights for the first layer in the group, and *mean-pooling* the base-model KV projection weights for all layers in the group. Similarly to Ainslie et al. [2023], we found in early experiments that mean-pooling performed best, and chose to employ mean-pooling in all further experiments. To account for the fact that different base-model layers have different RMSNorm parameters, we absorb the elementwise multiplication for each RMSNorm into the weights of its associated KV projection prior to mean-pooling, and initialize the RMSNorm multiplication parameters for all KV projections in the adapted model to 1.

**Results.** We found that our adapted CLA2 models quickly converged to low loss values, but did not fully recover the performance of their original base models after 21B tokens of adaptation training. We present the main results of our adaptation experiments in Table 5, and present training loss curves for the models in our adaptation experiments in Appendix H.2.

We believe it is likely possible to improve upon our CLA adaptation recipe, and leave doing so as a promising direction for future work.

| Model | Hellaswag | PIQA | WG | SciQ | OBQA | BoolQ | ARC-E | Wikitext Perplexity |
|-------|-----------|------|----|------|------|-------|-------|---------------------|
| 1B Base | **37.14** | **68.50** | 50.83 | **81.6** | 19.0 | **60.73** | **55.68** | **18.64** |
| 1B Adapt | 35.92 | 68.44 | **51.46** | 80.7 | **19.4** | 59.79 | 54.04 | 19.71 |
| 3B Base | **45.94** | 73.67 | **59.59** | **89.5** | **24.2** | 59.05 | **63.80** | **13.39** |
| 3B Adapt | 44.08 | **73.99** | 57.70 | 87.2 | 24.0 | **59.45** | 63.38 | 14.05 |

Table 5: Results for our MQA-to-CLA2 adaptation experiments.

# 5 Limitations & Future Work

For full-scale LLM serving systems, we expect KV cache reduction techniques like CLA to enable longer sequences, larger batch sizes, and greater KV cache persistence times than would otherwise be possible or economical. We do not directly evaluate these system-level consequences of reducing KV cache memory overhead, and instead leave end-to-end inference efficiency evaluations of large, long-context models employing CLA as an interesting problem for future work.

# 6 Related Work

**KV Cache Compression.**    Many works have tried to compress LLMs through pruning, quantization, and sparsity (see Zhu et al. [2024] for a survey), and a subset of these focus on KV cache compression. KVQuant [Hooper et al., 2024] and Coupled Quantization [Zhang et al., 2024b] employ targeted transformation of the keys and values, along with non-uniform numeric encodings, to compress the KV cache to 1-bit or 2-bit precision. Work on sparsifying the KV cache includes H2O [Zhang et al., 2023], Scissorhands [Liu et al., 2023], and FastGen [Ge et al., 2024], all of which employ heuristics to select unimportant tokens to discard. PagedAttention [Kwon et al., 2023] decreases the memory footprint of the KV cache in batched inference settings by eliminating unnecessary padding and allowing for sharing of common prefixes. Finally, Cachegen [Liu et al., 2024] directly losslessly compresses the KV cache with a custom-tailored arithmetic-coding-based compression scheme.

**KV-Cache-Efficient Transformer Variants.**    Significant prior work has proposed methods for modifying the transformer architecture in ways which reduce the size of the KV cache. Two notable early works in this direction are Transformer XL [Dai et al., 2019] and Sparse Attention [Child et al., 2019] which reduce effective sequence lengths by performing attention only within a local window. More recently, this line of work has been extended by methods like Infini-attention [Munkhdalai et al., 2024], which maintain fixed-sized compressive memories of tokens seen prior to a local window. Finally, MQA and GQA [Shazeer, 2019, Ainslie et al., 2023] enable substantial reductions in KV cache size, and serve as the direct inspiration for our technique.

**Removing Softmax Attention.**    A large body of recent work has proposed efficient alternatives to softmax attention for performing sequence-mixing in LLMs; notable ongoing work in this direction includes variants of linear attention [Katharopoulos et al., 2020, Wang et al., 2020, Yang et al., 2024], state space models (SSMs) such as Mamba [Gu and Dao, 2024], and the RWKV v6 architecture [Peng et al., 2024]. While these non-transformer architectures are often much more memory-efficient at inference time than a transformer employing a KV cache would be, characterizing their exact capabilities and limitations relative to transformer models remains an open area of research.

# 7 Conclusion

In this work, we introduce Cross-Layer Attention (CLA), a new attention architecture for transformer-based language models which reduces the size of the KV cache by sharing KV activations across layers. Through extensive experiments, we demonstrate that CLA allows more favorable accuracy/memory tradeoffs than are possible with MQA and GQA alone, thereby advancing the accuracy/memory Pareto frontier. In particular, we show that when combined with MQA with typical head dimensions, CLA allows $2\times$ reductions in KV cache size with minimal impact on accuracy, resulting in a simple and effective recipe for practitioners to improve the memory efficiency of their transformer models.

## 8 Funding Disclosure

Support and computational resources for this work were provided by the MIT-IBM Watson AI Lab. The authors declare no competing interest.

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

# A  Cross-Layer Attention Architecture

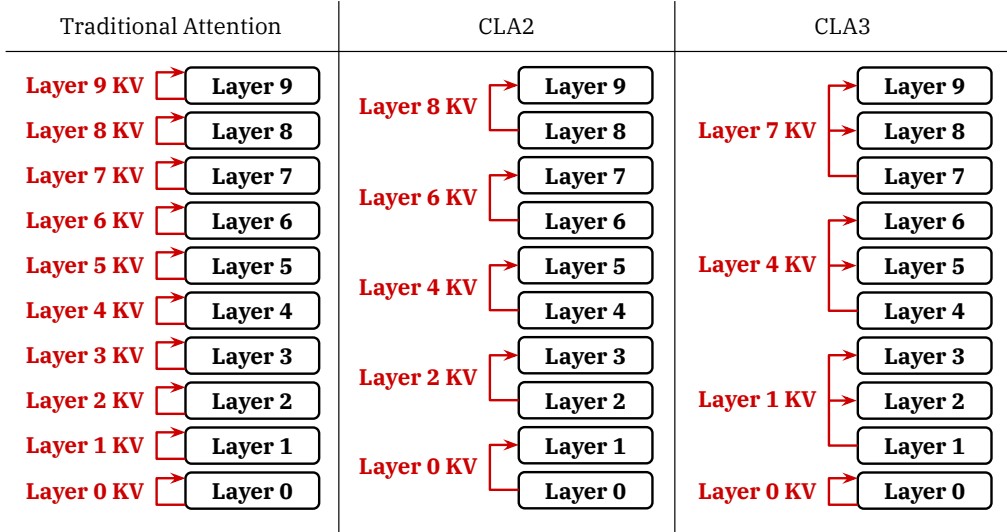

Figure 4: Schematic of KV cache structures under different attention configurations in a 10-layer transformer. Using traditional attention, each layer has its own KV cache. Using Cross-Layer Attention with a sharing factor of 2 (CLA2), every group of 2 consecutive layers shares a single KV cache. Using Cross-Layer Attention with a sharing factor of 3 (CLA3), every group of 3 consecutive layers shares a single KV cache. When the sharing factor does not evenly divide the number of layers, as in the CLA3 example, some KV caches must be shared over fewer layers than others; in this CLA3 configuration, we arbitrarily select the layer 0 KV cache to be used only in layer 0.

# B 1B-Scale Ablations

| Model | $d_{\text{head}}$ | Query Heads | KV Heads | KV Layers | KV Bytes Per Token (16-Bit) | Validation Perplexity |
|---|---|---|---|---|---|---|
| **Non-CLA Baselines** | | | | | | |
| H128-MHA | 128 | 16 | 16 | 20 | 163 840 | 13.15 |
| H128-GQA4 | 128 | 16 | 4 | 20 | 40 960 | 13.36 |
| H128-GQA2 | 128 | 16 | 2 | 20 | 20 480 | 13.52 |
| H128-MQA | 128 | 16 | 1 | 20 | 10 240 | 13.54 |
| H64-MQA | 64 | 32 | 1 | 20 | 5120 | 13.81 |
| H46-MQA | 46 | 45 | 1 | 20 | 3680 | 13.96 |
| H32-MQA | 32 | 64 | 1 | 20 | 2560 | 14.37 |
| **MQA + CLA2 Models** | | | | | | |
| H512-MQA-CLA2 | 512 | 4 | 1 | 10 | 20 480 | 13.49 |
| H256-MQA-CLA2 | 256 | 8 | 1 | 10 | 10 240 | 13.51 |
| H128-MQA-CLA2 | 128 | 16 | 1 | 10 | 5120 | 13.60 |
| H90-MQA-CLA2 | 90 | 22 | 1 | 10 | 3600 | 13.73 |
| H64-MQA-CLA2 | 64 | 32 | 1 | 10 | 2560 | 13.89 |
| **GQA + CLA2 Models** | | | | | | |
| H256-GQA4-CLA2 | 256 | 8 | 4 | 10 | 40 960 | 13.38 |
| H128-GQA4-CLA2 | 128 | 16 | 4 | 10 | 20 480 | 13.48 |
| H128-GQA2-CLA2 | 128 | 16 | 2 | 10 | 10 240 | 13.59 |
| **MQA + CLA > 2 Models** | | | | | | |
| H128-MQA-CLA3 | 128 | 16 | 1 | 7 | 3584 | 13.77 |
| H128-MQA-CLA4 | 128 | 16 | 1 | 5 | 2560 | 13.95 |
| **MQA + CLA2, Non-Uniform Sharing** | | | | | | |
| H128-MQA-CLA2-KeepEnds | 128 | 16 | 1 | 11 | 5632 | 13.62 |
| H128-MQA-CLA2-DenseFront | 128 | 16 | 1 | 11 | 5632 | 13.75 |
| H128-MQA-CLA2-DenseBack | 128 | 16 | 1 | 11 | 5632 | 14.03 |

Table 6: Full results of our 1B-scale design space exploration.

We conducted a number of ablations as part of our design space exploration at the 1B-parameter scale. The full results of our design space exploration, including ablations, are presented in Table 6. In this appendix, we describe the configurations and results for those ablations in more detail.

**Ablation: GQA + CLA2.** We trained three models to explore combining GQA with CLA2. We chose GQA4-CLA2 with $d_{\text{head}} = 128$ as our starting point, as GQA4 represents an attention configuration intermediate between our MQA and MHA baselines. We then explored expanding the head dimension of our GQA4-CLA2 model to $d_{\text{head}} = 256$, as well as reducing the GQA factor to GQA2. We found that only the GQA2-CLA2 configuration was able to achieve a perplexity better than the corresponding baseline model with the same KV cache footprint, and that this perplexity was the same (within 0.01 points) as our MQA-CLA2 model with the same footprint.

**Ablation: MQA + CLA with Sharing Factor > 2.** To explore the effect of using CLA sharing factors > 2, we trained MQA-CLA3 and MQA-CLA4 models with head dimension $d_{\text{head}} = 128$. We found that these CLA3 and CLA4 models achieved a Pareto improvement over our plain MQA baselines, matching the KV cache footprint of our baseline MQA models with head dimensions of $d_{\text{head}} \in \{32, 46\}$ while achieving better perplexities. However, we found that they achieved worse perplexities than our MQA-CLA2 models at the same KV cache footprint.

**Ablation: MQA + CLA2 with Non-Uniform Sharing Patterns.** Finally, we explored using different patterns of KV activation sharing in our MQA-CLA2 models.

On the hypothesis that the first and last layers in the model might benefit from special treatment, we trained a model "H128-MQA-CLA2-KeepEnds" which does not share the layer 0 KV cache with any other layers, and instead groups layer 1 with layer 2, groups layer 3 with layer 4, and so on. This also has the effect of giving the final layer its own KV cache separate from all other layers.

We also explored imbalanced configurations with all the KV-cache-producing layers concentrated at either the beginning or end of the model. We trained a model "H128-MQA-CLA2-DenseFront" consisting of 10 non-CLA layers, followed by 9 CLA layers all using the KV activations of layer 9, and a final layer with its own KV cache. Similarly, we trained a model "H128-MQA-CLA2-DenseBack" consisting of 2 non-CLA layers, followed by a run of 10 CLA layers all using the KV activations of layer 1, and finally 9 non-CLA layers.

We found that all of these alternative CLA sharing patterns achieve worse perplexities than the corresponding MQA-CLA2 model with a uniform sharing pattern, while also requiring slightly more KV cache memory.

# C  Full Results for Design Space Exploration

Table 7: Full benchmarking results for models in our 1B-Scale design space exploration.

| Model | ↑ hellaswag | ↑ piqa | ↑ winogrande | ↑ sciq | ↑ openbookqa | ↑ boolq | ↑ arc-e | ↓ wikitext (PPL) |
|---|---|---|---|---|---|---|---|---|
| H128-MHA | 33.88 | 67.19 | 53.12 | 81.1 | 19.0 | 61.62 | 52.15 | 20.90 |
| H128-GQA4 | 33.82 | 67.79 | 51.62 | 78.6 | 18.6 | 60.73 | 51.47 | 21.38 |
| H128-GQA2 | 33.34 | 67.85 | 53.04 | 79.6 | 20.0 | 60.89 | 50.97 | 21.64 |
| H128-MQA | 33.53 | 67.79 | 51.07 | 78.4 | 18.6 | 56.61 | 51.35 | 21.79 |
| H64-MQA | 33.24 | 67.52 | 50.04 | 75.8 | 17.0 | 59.39 | 51.22 | 22.31 |
| H46-MQA | 32.99 | 66.70 | 52.41 | 77.9 | 19.2 | 60.18 | 49.34 | 22.59 |
| H32-MQA | 32.58 | 67.80 | 50.99 | 74.5 | 18.4 | 59.94 | 49.02 | 23.76 |
| H512-MQA-CLA2 | 33.68 | 67.68 | 52.33 | 77.3 | 18.8 | 55.72 | 52.19 | 22.42 |
| H256-MQA-CLA2 | 33.90 | 67.74 | 49.80 | 77.4 | 18.2 | 60.34 | 50.04 | 21.64 |
| H128-MQA-CLA2 | 33.29 | 67.63 | 49.88 | 78.3 | 18.0 | 59.51 | 49.62 | 21.82 |
| H90-MQA-CLA2 | 33.15 | 67.41 | 51.85 | 76.7 | 17.2 | 59.11 | 52.06 | 22.13 |
| H64-MQA-CLA2 | 32.71 | 67.36 | 51.70 | 74.9 | 19.4 | 54.68 | 50.88 | 22.43 |
| H256-GQA4-CLA2 | 33.63 | 66.92 | 51.78 | 78.5 | 18.6 | 60.43 | 51.18 | 21.40 |
| H128-GQA4-CLA2 | 33.64 | 67.74 | 50.59 | 78.1 | 18.6 | 58.78 | 51.09 | 21.66 |
| H128-GQA2-CLA2 | 33.39 | 67.14 | 52.17 | 77.3 | 19.8 | 59.45 | 51.26 | 21.83 |
| H128-MQA-CLA3 | 32.91 | 67.74 | 51.54 | 76.6 | 18.0 | 54.53 | 51.18 | 22.18 |
| H128-MQA-CLA4 | 32.51 | 67.57 | 51.85 | 75.4 | 18.6 | 59.33 | 51.73 | 22.62 |
| H128-MQA-CLA2-KeepEnds | 33.58 | 68.12 | 52.72 | 76.2 | 19.2 | 60.12 | 51.64 | 21.88 |
| H128-MQA-CLA2-DenseFront | 33.43 | 67.30 | 52.57 | 75.7 | 19.4 | 49.14 | 50.88 | 22.09 |
| H128-MQA-CLA2-DenseBack | 32.71 | 66.65 | 51.70 | 76.5 | 17.4 | 59.69 | 50.51 | 22.80 |

# D  1B-Scale Learning Rate Sweeps

Here we present the results of our learning rate sweeps at the 1B-parameter scale:

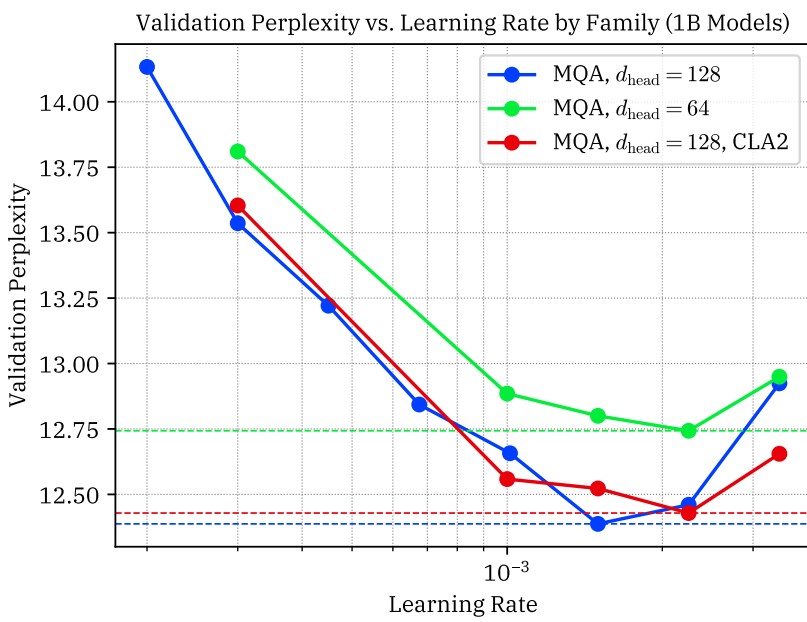

Figure 5: Results of our learning rate sweep at 1B-parameter scale. Dashed lines indicate the best loss achieved by each model family. We find that our MQA-CLA2 model and our MQA baseline with $d_{\text{head}} = 64$ both benefit from a higher learning rate than our MQA baseline with $d_{\text{head}} = 128$.

# E  Initial 3B-Scale Experiments

| Model | KV Bytes Per Token (16-bit) | Best LR | Validation Perplexity | Wikitext Perplexity |
|---|---|---|---|---|
| H128-MQA | 16384 | $6.75 \times 10^{-4}$ | 9.52 | 13.63 |
| H128-MQA-CLA2 | 8192 | $2.25 \times 10^{-3}$ | **9.34** | **13.25** |
| H64-MQA | 8192 | $1.00 \times 10^{-3}$ | 9.48 | 13.49 |

Table 8: Optimal learning rate and perplexity results for our initial 3B-scale experiments.

| Model (Best LR) | Hellaswag | PIQA | WG | SciQ | OBQA | BoolQ | ARC-E |
|---|---|---|---|---|---|---|---|
| H128-MQA | 45.73 | 73.07 | 60.46 | 88.1 | 25.4 | 59.30 | **64.90** |
| H128-MQA-CLA2 | **47.12** | **74.32** | **60.69** | **89.2** | 25.2 | 58.62 | 64.73 |
| H64-MQA | 46.42 | 74.05 | 57.85 | 88.1 | **25.6** | **59.88** | 65.57 |

Table 9: Downstream evaluation results for our initial 3B-scale experiments.

**Experiments at Head Dimension $d_{\text{head}} = 128$.**  We initially ran experiments to compare three 3B-scale models analogous to the models we selected for our learning rate tuning experiments at the 1B-parameter scale. Specifically, we compared a model using MQA-CLA2 and $d_{\text{head}} = 128$ to an MQA model with the same head dimension (and hence $2\times$ the KV cache footprint), and to an MQA model with a head dimension of $d_{\text{head}} = 64$ (and hence the same KV cache footprint). Based on our 1B-scale experiments, we expected that our MQA-CLA2 and MQA models with $d_{\text{head}} = 128$ would achieve similar perplexities to each other, and that both would outperform the $d_{\text{head}} = 64$ model.

We tuned the learning rates for these models according to the same learning rate tuning protocol we used at the 1B-parameter scale. After tuning the learning rates for each model, we observed a result different than we had expected: at 3B scale, our MQA-CLA2 model achieves substantially better perplexities than both our $d_{\text{head}} = 128$ and $d_{\text{head}} = 64$ MQA baselines. Moreover, our $d_{\text{head}} = 64$ MQA baseline model achieves better perplexities than our tuned $d_{\text{head}} = 128$ MQA baseline, despite having only $1/2$ as much KV cache capacity. We report the optimal learning rates and perplexities for these three models in Table 8, and present the results of our learning rate sweep graphically in Figure 6.

As with our 1B-scale learning rate tuning experiments, we evaluate these models on downstream benchmarks. We report the results of these evaluations in Table 9. As with our 1B-scale experiments, we do not find that any model consistently wins or loses in these downstream evaluations.

Due to logistical constraints, we used a different training cluster, software stack, and data order for our initial 3B experiments than for the main 3B experiments we describe in Section 3.3. To control for differences in training environment, our main 3B results reported in Section 3.3 use a separately-trained version of our H64-MQA-CLA2 baseline, distinct from the H64-MQA-CLA2 model described in this appendix.

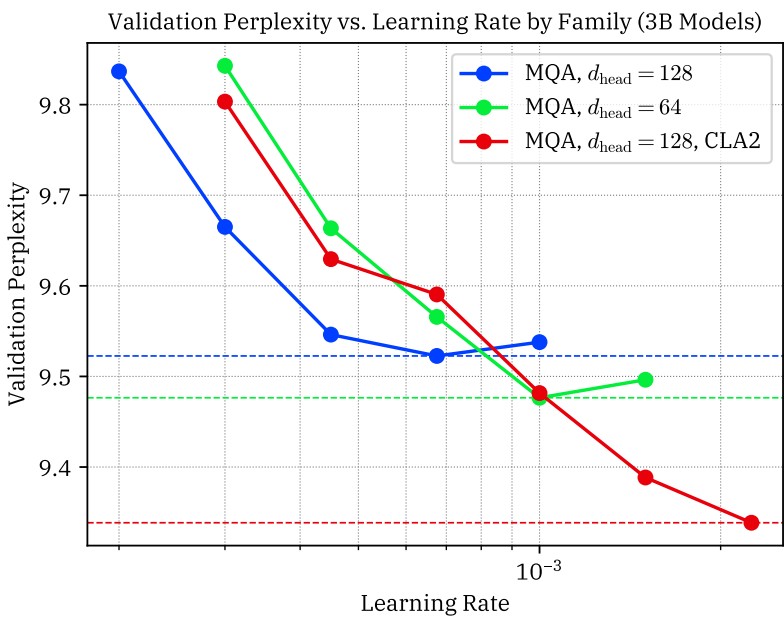

Figure 6: Learning rate sweep results for the model families in our initial 3B-scale experiments. Dashed lines indicate the best loss achieved by each model family. We find that our MQA-CLA2 model benefits from a higher learning rate than our $d_{\text{head}} = 64$ MQA model, which benefits from a higher learning rate than our $d_{\text{head}} = 128$ MQA model. At their best learning rates, we find that our MQA-CLA2 model outperforms our $d_{\text{head}} = 64$ MQA model, which outperforms our $d_{\text{head}} = 128$ model. We tried training our MQA-CLA2 model at LR $= 3.375 \times 10^{-3}$ – one LR increment higher than the rightmost point depicted in this plot – but found that training diverged at that LR scale.

# F Training Loss Curves

Here we present visualizations of the training loss curves for the models in our main experiments at the 1B- and 3B-parameter scales. The data in each plot has been smoothed using an exponential moving average, with the same smoothing factor used for all plots.

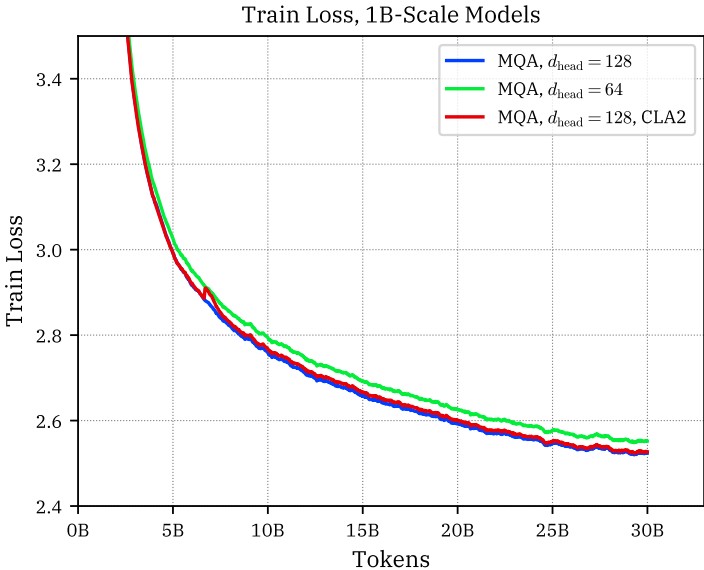

Figure 7: Training loss curves for the 1B-scale models described in Section 3.2.2, each at its best learning rate.

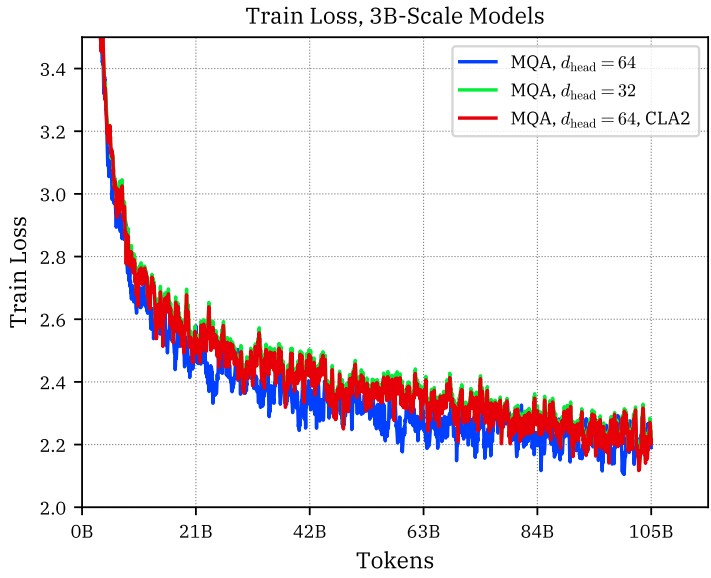

Figure 8: Training loss curves for the models from our main 3B-scale experiments, described in Section 3.3. Due to a logging bug, training loss was reported based on the examples seen by only a single device on each training step, rather than being averaged over the entire data-parallel group. This bug resulted in noisier loss measurements being logged (but had no effect on training dynamics).

# G    Comparison to Open Model – Results

| Model | Hellaswag | PIQA | WG | SciQ | OBQA | BoolQ | ARC-E | Wikitext Perplexity |
|---|---|---|---|---|---|---|---|---|
| Original | 43.54 | 67.30 | 53.67 | 72.40 | 29.80 | **59.63** | 44.91 | 21.34 |
| CLA2 | **45.83** | **68.55** | **53.75** | **77.30** | **32.60** | 59.30 | **47.10** | **19.59** |

Table 10: Downstream benchmarks results for the publicly-available TinyLlama-1.1B-105B checkpoint, and our version of it trained from scratch with CLA2.

# H    Adaptation Experiments

## H.1    Hyperparameters

The models trained in the adaptation experiments described in Section 4 used the architectural hyperparameters given in Table 11.

| Model | Hidden Size | FFN Size | Layers | Sequence Length | Attention Mechanism | Attention Head Size | Initialization |
|---|---|---|---|---|---|---|---|
| 1B Base | 2048 | 5472 | 20 | 2048 | MQA | 64 | Random |
| 1B Adapt | 2048 | 5472 | 20 | 2048 | MQA-CLA2 | 64 | From 1B Base |
| 3B Base | 3072 | 8192 | 32 | 2048 | MQA | 256 | Random |
| 3B Adapt | 3072 | 8192 | 32 | 2048 | MQA-CLA2 | 256 | From 3B Base |

Table 11: Architectural hyperparameters for models used in adaptation experiments.

We used the same training data and tokenizer in our adaptation experiments as in our pretraining experiments. The training hyperparameters for each model are given in Table 12. Our cosine learning rate schedules always decay to $10\%$ of the peak LR and reach the end of their cycle at exactly the end of training.

| Model | Training Tokens | LR Schedule | LR Warmup Tokens | Peak LR |
|---|---|---|---|---|
| 1B Base | $105 \times 10^9$ | Warmup + Cosine | $5 \times 10^9$ | $3.00 \times 10^{-4}$ |
| 1B Adapt | $21 \times 10^9$ | Warmup + Cosine | $4 \times 10^9$ | $3.00 \times 10^{-4}$ |
| 3B Base | $105 \times 10^9$ | Warmup + Cosine | $5 \times 10^9$ | $6.75 \times 10^{-4}$ |
| 3B Adapt | $21 \times 10^9$ | Warmup + Cosine | $4 \times 10^9$ | $3.00 \times 10^{-4}$ |

Table 12: Training hyperparameters for models used in adaptation experiments.

## H.2  Training Loss Curves

Here, we present the training loss curves for the models in our 1B- and 3B-scale adaptation experiments, described in Section 4. In each experiment, we first pretrained an MQA "base" model (indicated in blue) on 105B tokens, and then used that model's final weights to initialize an MQA-CLA2 "adapted" model (indicated in red) trained on a further 21B tokens.

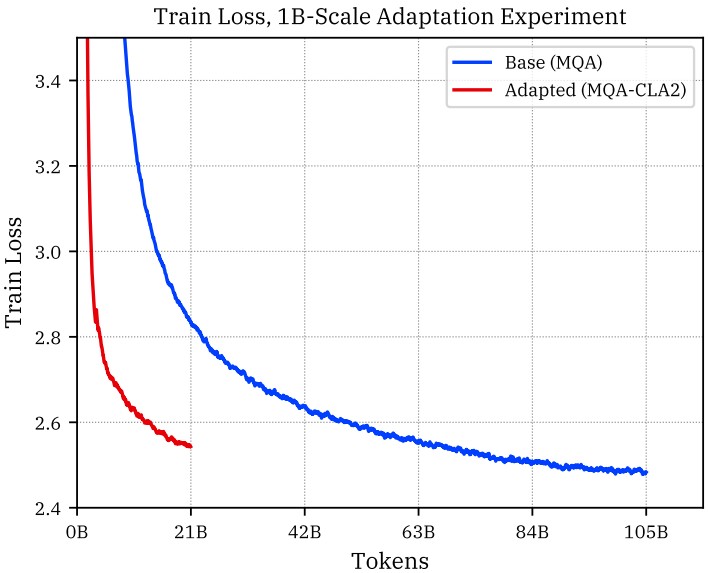

Figure 9: Training loss curves for our 1B-scale adaptation experiment.

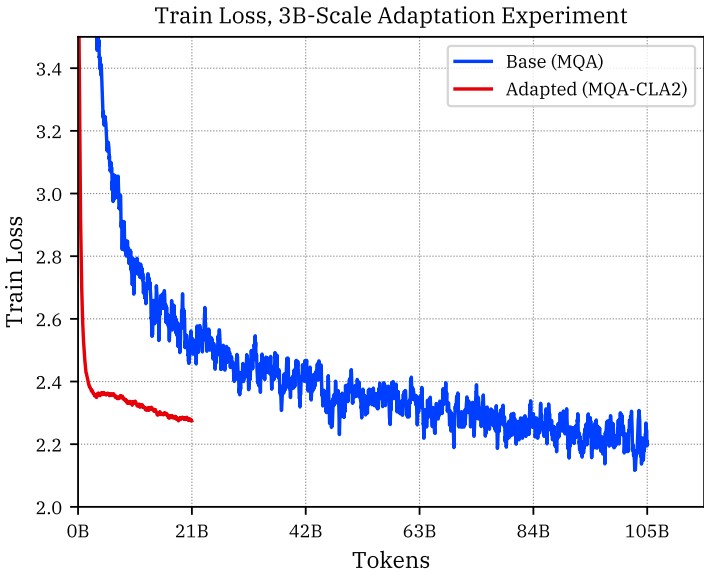

Figure 10: Training loss curves for our 3B-scale adaptation experiment. Due to a logging bug, training loss for the base model was reported based on the examples seen by only a single device on each training step, rather than being averaged over the entire data-parallel group. This bug resulted in noisier loss measurements being logged (but had no effect on training dynamics).

# I  Information About Assets Used

We used the following software assets in conducting the experiments for this paper:

- PyTorch version 2.1.2, made available under the BSD-3 license.
  - https://pytorch.org
- SlimPajama, made available under the Apache 2.0 license.
  - https://www.cerebras.net/blog/slimpajama-a-627b-token-cleaned-and-deduplicated-version-of-redpajama

