# OpenReview forum: "Reducing Transformer Key-Value Cache Size with Cross-Layer Attention"
_NeurIPS.cc/2024/Conference — NeurIPS 2024 poster_

### Official Review · Reviewer_mqB5 · 2024-07-13

**Soundness:** 3
**Presentation:** 2
**Contribution:** 2
**Rating:** 6
**Confidence:** 3

**Summary:**

To resolve the costly memory consumption of KV cache in transformer-based LLMs, this paper proposes Cross-Layer Attention (CLA). In contrast to multi-query attention (MQA) and grouped-query attention (GQA) which share KV cache across attention heads, CLA shares KV cache across contiguous layers. Pre-training experiments prove the effectiveness of CLA over MQA and GQA.

**Strengths:**

1. The experiments are comprehensive and the design choices are clearly documented.
2. The problem studied is an important one and the approach is novel.

**Weaknesses:**

1. Experimental results are not presented fully. Figure 3 only shows the validation perplexity for a subset of models, and Table 3 and 4 only presents the accuracy for a subset of models. I couldn't find the results elsewhere in the paper or the appendix. Presenting the results fully contributes to the holistic understanding of the effectiveness of the proposed approach.
2. Suggestions to experimental designs. The authors choose to pre-train 1B and 3B models from scratch to prove the effectiveness of CLA. It would perhaps be more convincing if the authors can choose a fully open source pre-trained model (such as OpenLLaMA) as baseline, and pre-train it with CLA to prove its effectiveness. Pre-training 1B and 3B models using a custom set of hyper-parameters and datasets is less convincing in comparison.
3. Training loss curves are missing. Training loss is important for understanding whether the models have fully converged.
4. Citation formats may be incorrect. Many references on page 10-12 only show author names, title, and year, and omit publication venue.
5. Some typos. Line 16: "which are possible" -> "which are impossible", line 18: "would otherwise be possible" -> "would otherwise be impossible".

**Questions:**

1. Can the authors please provide some insights on why cross-layer sharing of KV cache may produce better models than GQA? The proposed method would be more convincing if some design justifications are provided.
2. Is it possible to adapt pre-trained models to using CLA through fine-tuning?

**Limitations:**

Yes the authors addressed the limitations. I do not envision additional limitations.

---

> ### Author Rebuttal · Authors · 2024-08-07
>
> We thank the reviewer for their thoughtful and generous feedback. We address each point below:
>
> ## Could we include more comprehensive results tables in the paper?
>
> We thank the reviewer for pointing this out, and agree the paper could be improved by including more comprehensive results tables.
>
> For validation perplexities, we note that validation perplexities for all model architectures in our design space exploration, including those not shown in Table 1 or Figure 3, are already included in Appendix B Table 5.
>
> For accuracies on downstream tasks, the reviewer is correct that our initial submission only included downstream accuracies for a subset of architectures. Although we include downstream accuracies for selected 1B-scale architectures (see Table 3) and all 3B architectures (see Table 4 and Appendix D Table 7), we do not include downstream accuracy results for all architectures in our 1B-scale design space exploration. We will correct this in the final submission, and include a table (likely in the appendix) containing the missing benchmark results for all architectures not currently documented. We have provided this full table of results in an accompanying comment on OpenReview.
>
> ## Could we compare to an open-source pre-trained model?
>
> We thank the reviewer for their suggestion. In response to the reviewer's comment, we have run an experiment comparing directly with the open-source GQA4 model TinyLlama-1.1B at its 105B-token intermediate checkpoint. In this experiment, we pretrained our own version of TinyLlama-1.1B-105B from scratch using CLA2, using otherwise-identical training data, model architecture, and training hyperparameters as described in the TinyLlama repository. In particular, we used the same cosine learning rate schedule as TinyLlama 1.1B, which decays over 3T tokens (although we only ran training to 105B tokens).
>
> In this comparison, we find that our TinyLlama-1.1B-105B checkpoint trained with CLA2 matches or exceeds the performance of the original TinyLlama-1.1B-105B checkpoint. We include the benchmark scores for each model below:
>
> | Model              | $\uparrow$ hellaswag | $\uparrow$ piqa | $\uparrow$ winogrande | $\uparrow$ sciq | $\uparrow$ openbookqa | $\uparrow$ boolq | $\uparrow$ arc-e | $\downarrow$ Wikitext (ppl)|
> |--------------------|----------------------|-----------------|-----------------------|-----------------|-----------------------|------------------|---------------------|-----------------------|
> | Tiny-Llama-CLA     | 0.4583               | 0.6855          | 0.5375                | 0.7730          | 0.3260                | 0.5930           | 0.4710              | 19.5853               |
> | Tiny-Llama-Original| 0.4354               | 0.6730          | 0.5367                | 0.7240          | 0.2980                | 0.5963           | 0.4491              | 21.3407               |
>
> At the reviewers' discretion, we would be happy to include these results in the final version of our paper.
>
> ## Could we include training loss curves?
>
> We would be happy to extend the appendix in the final version of our paper to include visualizations of the training loss curves for all the models in our experiments. Although we are not able to attach images of the loss curves in this OpenReview response, we can say that the shapes of the loss curves for all CLA and non-CLA models are qualitatively similar, and that all models have loss curves typical of transformer-based LLMs trained with cosine-decay learning rate schedules.
>
> ## Formatting and typos
>
> We appreciate the careful attention the reviewer put into our work, and will fix these errors.
>
> ## Why should we expect cross-layer sharing to help relative to using only GQA/MQA?
>
> GQA and MQA can be seen as relying on the hypothesis that in MHA models there is hidden redundancy in the content of KV activations in _different heads_ within each layer. GQA and MQA then exploit this hypothesized redundancy to reduce the size of the KV cache relative to MHA, with only minor degradation in performance.
>
> Similarly, our original motivation for investigating Cross-Layer Attention was based on the hypothesis that even with GQA/MQA, there may be remaining hidden redundancy in the content of KV activations across _different nearby layers_. If there is hidden redundancy across layers, then GQA and MQA have no mechanism for exploiting that redundancy to reduce KV cache size, no matter how we set our GQA/MQA hyperparameters -- however, CLA would be able to exploit that redundancy.
>
> We would be happy to mention this design motivation in the final version of our paper.
>
> ## What about adapting pre-trained models to use CLA?
>
> We agree that adapting (or "uptraining," as the GQA paper calls it) existing models to use CLA is an interesting avenue of research.
>
> We have conducted some preliminary experiments on CLA uptraining. We have found that it is possible to convert 1B- and 3B-scale MQA models each trained on 100B tokens into MQA-CLA2 models with 5.7% and 4.9% higher perplexity, respectively, by uptraining them with CLA for 20B tokens. Similarly to the GQA paper, we find that initializing the uptrained model by mean-pooling KV projection weights outperforms simply dropping KV projections from the model.
>
> At the reviewers' discretion, we would be happy to include these preliminary uptraining results in the final version of the paper. We also believe it is likely possible to improve further upon our uptraining scheme, and would be happy to mention in the paper that improved schemes for CLA uptraining represent a promising direction for future research.
>
> We also note that although the original GQA paper focused on uptraining, GQA has also had a significant impact on industry practice via its direct application to LLM pretraining, as seen in models like the Llama 3 series. We hope that CLA may be able to have a similar impact via direct application to pretraining.

---

> ### Author Response · Authors · 2024-08-07
> **Full Benchmarking Results for Models in our Design Space Exploration**
>
> Attached please find the full benchmarking results for the models in our 1B-scale design space exploration. (This comment accompanies our rebuttal, which could not fit the attached table due to length constraints.)
>
> |Model|$\uparrow$ hellaswag|$\uparrow$ piqa|$\uparrow$ winogrande|$\uparrow$ sciq|$\uparrow$ openbookqa|$\uparrow$ boolq|$\uparrow$ arc-e|$\downarrow$ wikitext (PPL)|
> |-|-|-|-|-|-|-|-|-|
> |H128-MHA|33.88|67.19|53.12|81.1|19|61.62|52.15|20.90|
> |H128-GQA4|33.82|67.79|51.62|78.6|18.6|60.73|51.47|21.38|
> |H128-GQA2|33.34|67.85|53.04|79.6|20|60.89|50.97|21.64|
> |H128-MQA|33.53|67.79|51.07|78.4|18.6|56.61|51.35|21.79|
> |H64-MQA|33.24|67.52|50.04|75.8|17|59.39|51.22|22.31|
> |H46-MQA|32.99|66.7|52.41|77.9|19.2|60.18|49.34|22.59|
> |H32-MQA|32.58|67.8|50.99|74.5|18.4|59.94|49.02|23.76|
> |H512-MQA-CLA2|33.68|67.68|52.33|77.3|18.8|55.72|52.19|22.42|
> |H256-MQA-CLA2|33.9|67.74|49.8|77.4|18.2|60.34|50.04|21.64|
> |H128-MQA-CLA2|33.29|67.63|49.88|78.3|18|59.51|49.62|21.82|
> |H90-MQA-CLA2|33.15|67.41|51.85|76.7|17.2|59.11|52.06|22.13|
> |H64-MQA-CLA2|32.71|67.36|51.7|74.9|19.4|54.68|50.88|22.43|
> |H256-GQA4-CLA2|33.63|66.92|51.78|78.5|18.6|60.43|51.18|21.40|
> |H128-GQA4-CLA2|33.64|67.74|50.59|78.1|18.6|58.78|51.09|21.66|
> |H128-GQA2-CLA2|33.39|67.14|52.17|77.3|19.8|59.45|51.26|21.83|
> |H128-MQA-CLA3|32.91|67.74|51.54|76.6|18|54.53|51.18|22.18|
> |H128-MQA-CLA4|32.51|67.57|51.85|75.4|18.6|59.33|51.73|22.62|
> |H128-MQA-CLA2-KeepEnds|33.58|68.12|52.72|76.2|19.2|60.12|51.64|21.88|
> |H128-MQA-CLA2-DenseFront|33.43|67.3|52.57|75.7|19.4|49.14|50.88|22.09|
> |H128-MQA-CLA2-DenseBack|32.71|66.65|51.7|76.5|17.4|59.69|50.51|22.8|

---

> > ### Comment · Reviewer_mqB5 · 2024-08-08
> >
> > Thank you to the authors for the detailed response. My concerns have mostly been addressed. I think the additional clarifications and experimental results are a great addition to the paper. I have adjusted my initial rating accordingly.

---

> > > ### Author Response · Authors · 2024-08-08
> > > **Thank You**
> > >
> > > We are glad that we were able to address the reviewer's questions, and thank them for their response.

---

### Official Review · Reviewer_skpH · 2024-07-14

**Soundness:** 3
**Presentation:** 3
**Contribution:** 3
**Rating:** 7
**Confidence:** 4

**Summary:**

In this paper, the authors proposed an approach, Cross-Layer Attention (CLA), to accelerate the autoregressive generation procedure of Large Language Models. Going beyond multi-query attention, the main idea of CLA is to share the Key-Value caches among attention layers. Intuitively, this idea is straightforward to further reduce the size of the KV cache that must be stored during generation. The authors conducted several experiments covering different model sizes and combinations between CLA and GQA/MQA to verify the effectiveness of the CLA module.

**Strengths:**

1. The problem this paper aims to tackle has great significance in real-world applications of LLMs.
2. Overall, the proposed CLA approach is simple and easy to implement, which is friendly for practical engineering. The paper is also easy to follow.
3. **Efficiency**: from the experimental results, the CLA module can achieve further acceleration beyond MQA/GQA approaches across different model sizes.
4. **Performance**: it is interesting to see that the cross-layer sharing of KV caches does not hurt the performance of LLMs a lot, which further serves as support evidence that the CLA approach can be applied in practice.

**Weaknesses:**

1. **The comprehensiveness of experiments can be further improved.**
    - **Model sizes**: the current version of this paper only verifies the effectiveness of CLA on 1B and 3B Transformer language models. Although the experimental results demonstrate that CLA does not bring performance drop on real-world tasks for these two model sizes, it is also questionable whether modern LLMs with  >10 and > 100 billion parameters could still use this approach to achieve better accuracy-efficiency. After all, there exist many approaches that are only effective on small-scale architectures in the literature. It is also understandable that the computational resources are restrictions for conducting such verification experiments, but I think it would be a great plus for improving the quality of this work.
    - **Design choices**: in Table 1 and Appendix B, the authors listed a bunch of Non-CLA baselines and MQA/GQA+CLA2/CLA3/CLA4 models. However, CLA-only models are not compared. It would be questionable whether CLA can only be used with MQA/GQA together.

**Questions:**

1. In lines 139-141, could you explain why using separately-learnable affine layer-norm parameters for the KV projection blocks and Q blocks in attention?
2. Similar to Sec 3.2.2, how is the robustness of CLA against other hyper-parameters like batch size?

**Limitations:**

The authors discussed limitations of this work in Sec 4.

---

> ### Author Rebuttal · Authors · 2024-08-07
>
> We thank the reviewer for their thoughtful and generous feedback. We address each point below:
>
> ## Could we evaluate CLA on larger models?
>
> Unfortunately, we lack the resources to train 10B- and 100B-scale models from scratch using CLA. Even training a single model on the same parameter and data scale as Llama 1 7B would require $30\times$ the training compute of our largest 3B-scale training runs. We hope that future work will apply CLA to models at larger scale.
>
> ## Why don't we evaluate CLA in combination with MHA?
>
> We chose to spend our resource budget evaluating models combining CLA with GQA/MQA, rather than MHA, because we believe that most practitioners who would apply CLA would want to do so in combination with GQA/MQA.
>
> As we mention in Section 2.3, CLA reduces KV cache storage footprint, whereas GQA/MQA reduces both KV cache storage footprint *and* memory bandwidth requirements during decoding. Because MQA/GQA provides these dual benefits, we expect practitioners to prefer using MQA/GQA first to obtain KV cache size reductions over MHA, and to only apply CLA to obtain further reductions on top of GQA/MQA. We note that because of the efficiency benefits of GQA at inference time, many recent models, such as Llama 3 and Mixtral, use GQA rather than MHA.
>
> We chose to include the MHA model in our design space exploration just as a point of comparison, to contextualize the performance achieved by other GQA/MQA and CLA models. We would be happy to clarify this point, and the reasoning for why we did not directly combine CLA with MHA, in the final version of our paper.
>
> ## Why do we use separately-learnable affine parameters for the KV projection?
>
> We chose to use separately-learnable affine parameters in the layernorms for our KV projections because we did not want to arbitrarily privilege one of the several attention layers sharing each KV projection to also share its layernorm affine parameters, and because it was a convenient way to implement CLA within our framework. We note that using separately-learnable affine parameters increases the parameter counts of our models by less than 0.003% in all cases. Due to limited resources, we did not ablate this choice.
>
> ## Could we conduct sweeps over batch size, or other hyperparameters?
>
> We chose to use our available resources to sweep over learning rate because it is typically the hyperparameter to which final model performance is most sensitive. For our choice of batch size, Adam $\beta_1$ and $\beta_2$, and AdamW weight decay coefficient, all our experiments used the same standard values used for pretraining Llama 2. We agree that, in the absence of resource constraints, sweeping over batch size and other hyperparameters would be valuable, but we note that the optimal learning rate can depend on batch size; this means that sweeping over batch size while ensuring we were comparing models at their best learning rates would in fact require a prohibitively expensive joint sweep over both batch size and learning rate. We thank the reviewer for their suggestion, and leave more exhaustive ablations of different training hyperparameters as a possible direction for future work.

---

> ### Comment · Reviewer_skpH · 2024-08-10
>
> Thank you for your responses. Although I still think the potential of this work should be further evaluated by using models of larger sizes, I understand the limitations of computational resources. I choose to raise my rating to 7.

---

### Official Review · Reviewer_8ZcC · 2024-07-27

**Soundness:** 3
**Presentation:** 3
**Contribution:** 4
**Rating:** 8
**Confidence:** 5

**Summary:**

This paper introduces a new KV cache compression technique by sharing KV cache cross-layers called CLA. It can be integrated with most of the existing KV cache compression technique like MQA, GQA, and quantization. When applying over GQA, CLA can further reduces KV cache size by 2× while maintaining similar accuracy. Experiments with 1B- and 3B-parameter models show that CLA offers better memory/accuracy trade-offs, allowing models to handle longer sequences and larger batch sizes.

**Strengths:**

- Simple and effective idea. It introduces an orthogonal dimension to the existing KV Cache compression methods. It can easily be integrated with existing KV cache techniques, like MQA/GQA, quantization, token eviction.

- The writing and presentation is easy to follow

- The proposed Cross-Layer Attention (CLA) does not require custom implementation and easily be deployed for any end-devices.

**Weaknesses:**

I personally like this paper and vote for acceptance. However, I think the experimental design could be improved to make it more solid:

- Only Wikitext Perplexity is reported, which is a weak indicator of LLM ability. While I understand that LLM evaluation is challenging and time-consuming, I suggest the authors also report the MMLU score, even if only on a subset.

- Can CLA work under continuous pretraining scenarios like GQA? Namely, can a pretrained LLM be continuously tuned into a CLA-based model, even though the compression ratio might not be as high in this case? I ask this because pretraining LLMs is extremely expensive, as seen with models like Llama 3.1. It is risky and costly to train another open-sourced CLA model from scratch. If we can adapt existing models into CLA-based models, this paper could have a much greater impact.

- One of the main motivations for KV Cache compression is long-context inference. However, long-context tasks like few-shot learning or multi-doc QA are much harder than the "short" tasks. I understand that long-context evaluation is challenging for pretrained models due to lack of alignment. Still, I think the authors could show some passkey retrieval results, as this task does not require much alignments.

**Questions:**

Check Weaknesses point 2

**Limitations:**

No major limitations.

---

> ### Author Rebuttal · Authors · 2024-08-07
>
> We thank the reviewer for their thoughtful and generous feedback. We address each point below:
>
> ## Could we report more quality metrics?
>
> In addition to Wikitext perplexity, we also report accuracy scores on Hellaswag, PIQA, WinoGrande, OpenBookQA, BoolQ, and ARC-E, which can be found in Table 3 and Table 4. In aggregate, we find that these metrics show that our CLA models match the accuracy of our non-CLA baselines while using only half as much KV cache memory.
>
> In response to your question, we have evaluated our 1B- and 3B-parameter models on MMLU, but find that MMLU does not provide useful signal for comparing the quality of different models at our scale of training compute. In particular, we observe that all the models which we compare in Figure 1 score no better than chance (25% accuracy) on MMLU:
>
> | Model | MMLU Accuracy |
> |-|-|
> | H128-MQA 1B (LR-tuned) | 23.76% |
> | H64-MQA 1B (LR-tuned) | 23.15% |
> | H128-MQA-CLA2 1B (LR-tuned) | 24.28% |
> | H64-MQA 3B (LR-tuned) | 23.32% |
> | H32-MQA 3B (LR-tuned) | 23.24% |
> | H64-MQA-CLA2 3B (LR-tuned) | 24.96% |
>
> For a point of reference, even the publicly-available open-source models TinyLlama 1.1B and OpenLlama 3B, each of which was trained with $\approx 10\times$ the FLOPs of our largest 3B training runs, do not achieve accuracy significantly better than chance on MMLU:
>
> | Publicly-Available Model | MMLU Accuracy |
> |-|-|
> | TinyLlama 1.1B | 25.34% |
> | OpenLlama 3B | 23.52% |
>
> ## What about continuous pretraining?
>
> We agree that continuous pretraining (or "uptraining" as the GQA paper calls it) for applying CLA to existing models is an interesting avenue of research.
>
> We have conducted some preliminary experiments on CLA uptraining. We have found that it is possible to convert 1B- and 3B-scale MQA models each trained on 100B tokens into MQA-CLA2 models with 5.7% and 4.9% higher perplexity, respectively, by uptraining them with CLA for 20B tokens. Similarly to the GQA paper, we find that initializing the uptrained model by mean-pooling KV projection weights outperforms simply dropping KV projections from the model.
>
> At the reviewers' discretion, we would be happy to include these preliminary uptraining results in the final version of the paper. We also believe it is likely possible to improve further upon our uptraining scheme, and would be happy to mention in the paper that improved schemes for CLA uptraining represent a promising direction for future research.
>
> We also note that although the original GQA paper focused on uptraining, GQA has also had a significant impact on industry practice via its direct application to LLM pretraining, as seen in models like the Llama 3 series. We hope that CLA may be able to have a similar impact via direct application to pretraining.
>
> ## What about long-context tasks?
>
> Training a high-quality long-context model with or without CLA was not possible within our resource constraints. As we mention in Section 4, we leave larger-scale evaluations of long-context, aligned models employing CLA as an interesting problem for future work. We thank the reviewer for their suggestion, and agree that in such future work, passkey retrieval would be a worthwhile task to evaluate.

---

> > ### Comment · Reviewer_8ZcC · 2024-08-09
> >
> > Thank you for your clarification. I am satisfied with the response.
> >
> > Good Luck!

---

### Author Rebuttal · Authors · 2024-08-07

We thank all the reviewers for their comments. We address their main points below:

## Could we report more quality metrics?

In addition to Wikitext perplexity, we also report accuracy scores on Hellaswag, PIQA, WinoGrande, OpenBookQA, BoolQ, and ARC-E, which can be found in Table 3 and Table 4. In aggregate, we find that these metrics show that our CLA models match the accuracy of our non-CLA baselines while using only half as much KV cache memory.

In response to Reviewer 8ZcC's question, we have evaluated our 1B- and 3B-parameter models on MMLU, but find that MMLU does not provide useful signal for comparing the quality of different models at our scale of training compute, as models at this scale perform no better than chance. We provide more detail in our response to Reviewer 8ZcC.

## What about adapting pre-trained models to use CLA?

We agree with reviewers 8ZcC and mqB5 that adapting (or "uptraining," as the GQA paper calls it) existing models to use CLA is an interesting avenue of research.

We have conducted some preliminary experiments on CLA uptraining. We have found that it is possible to convert 1B- and 3B-scale MQA models each trained on 100B tokens into MQA-CLA2 models with 5.7% and 4.9% higher perplexity, respectively, by uptraining them with CLA for 20B tokens. Similarly to the GQA paper, we find that initializing the uptrained model by mean-pooling KV projection weights outperforms simply dropping KV projections from the model.

At the reviewers' discretion, we would be happy to include these preliminary uptraining results in the final version of the paper. We also believe it is likely possible to improve further upon our uptraining scheme, and would be happy to mention in the paper that improved schemes for CLA uptraining represent a promising direction for future research.

We also note that although the original GQA paper focused on uptraining, GQA has also had a significant impact on industry practice via its direct application to LLM pretraining, as seen in models like the Llama 3 series. We hope that CLA may be able to have a similar impact via direct application to pretraining.

## What about long-context tasks?

Training a high-quality long-context model with or without CLA was not possible within our resource constraints. As we mention in Section 4, we leave larger-scale evaluations of long-context, aligned models employing CLA as an interesting problem for future work.

## Could we evaluate CLA on larger models?

Unfortunately, we lack the resources to train 10B- and 100B-scale models from scratch using CLA, as suggested by Reviewer skpH. Even training a single model on the same parameter and data scale as Llama 1 7B would require $30\times$ the training compute of our largest 3B-scale training runs. We hope that future work will apply CLA to models at larger scale.

## Why don't we evaluate CLA in combination with MHA?

We chose to spend our resource budget evaluating models combining CLA with GQA/MQA, rather than MHA, because we believe that most practitioners who would apply CLA would want to do so in combination with GQA/MQA.

As we mention in Section 2.3, CLA reduces KV cache storage footprint, whereas GQA/MQA reduces both KV cache storage footprint *and* memory bandwidth requirements during decoding. Because MQA/GQA provides these dual benefits, we expect practitioners to prefer using MQA/GQA first to obtain KV cache size reductions over MHA, and to only apply CLA to obtain further reductions on top of GQA/MQA. We note that because of the efficiency benefits of GQA at inference time, many recent models, such as Llama 3 and Mixtral, use GQA rather than MHA.

We chose to include the MHA model in our design space exploration just as a point of comparison, to contextualize the performance achieved by other GQA/MQA and CLA models. We would be happy to clarify this point, and the reasoning for why we did not directly combine CLA with MHA, in the final version of our paper.

## Could we include more comprehensive results tables in the paper?

We thank Reviewer mqB5 for pointing out that the paper could be improved by including more comprehensive results tables.

For validation perplexities, we note that validation perplexities for all model architectures in our design space exploration, including those not shown in Table 1 or Figure 3, are already included in Appendix B Table 5.

For accuracies on downstream tasks, the reviewer is correct that our initial submission only included downstream accuracies for a subset of architectures. Although we include downstream accuracies for selected 1B-scale architectures (see Table 3) and all 3B architectures (see Table 4 and Appendix D Table 7), we do not include downstream accuracy results for all architectures in our 1B-scale design space exploration. We will correct this in the final submission, and include a table (likely in the appendix) containing the missing benchmark results for all architectures not currently documented. We provide this table in the supplementary PDF (attached).

## Could we compare to an open-source pre-trained model?

We thank Reviewer mqB5 for their suggestion. In response to the reviewer's comment, we have run an experiment comparing directly with the open-source GQA4 model TinyLlama-1.1B at its 105B-token intermediate checkpoint. In this experiment, we pretrained our own version of TinyLlama-1.1B-105B from scratch using CLA2, using otherwise-identical training data, model architecture, and training hyperparameters as described in the TinyLlama repository. In particular, we used the same cosine learning rate schedule as TinyLlama 1.1B, which decays over 3T tokens (although we only ran training to 105B tokens).

In this comparison, we find that our TinyLlama-1.1B-105B checkpoint trained with CLA2 matches or exceeds the performance of the original TinyLlama-1.1B-105B checkpoint. We present these results in more detail in the supplementary PDF (attached), and in our response to Reviewer mqB5.

---

### Decision · Program_Chairs · 2024-09-25

**Decision:**

Accept (poster)

**Comment:**

The paper introduces a novel yet simple method for KV cache compression, one of the most challenging topics in today's deployment of transformer-like architectures. All reviews were positive, and the authors engaged in a fruitful discussion during rebuttal.
I think this result will benefit NeurIPS community.